# Investigation on the Adsorption-Interaction Mechanism of Pb(II) at Surface of Silk Fibroin Protein-Derived Hybrid Nanoflower Adsorbent

**DOI:** 10.3390/ma13051241

**Published:** 2020-03-09

**Authors:** Xiang Li, Yan Xiong, Ming Duan, Haiqin Wan, Jun Li, Can Zhang, Sha Qin, Shenwen Fang, Run Zhang

**Affiliations:** 1School of Chemistry and Chemical Engineering, Southwest Petroleum University, Chengdu 610500, China; 201721000245@stu.swpu.edu.cn (X.L.); 201821000253@stu.swpu.edu.cn (J.L.); 201922000213@stu.swpu.edu.cn (C.Z.); 201921000230@stu.swpu.edu.cn (S.Q.); mduana124@swpu.edu.cn (S.F.); 2State Key Laboratory of Pollution Control and Resource Reuse, Jiangsu Key Laboratory of Vehicle Emissions Control, School of the Environment, Nanjing University, Nanjing 210023, China; wanhq@nju.edu.cn; 3Australian Institute for Bioengineering and Nanotechnology, AIBN, The University of Queensland, St Lucia, QLD 4072, Australia; r.zhang@uq.edu.au

**Keywords:** silk fibroin, hybrid nanoflowers surface, Pb(II) removal, interaction mechanism

## Abstract

For further the understanding of the adsorption mechanism of heavy metal ions on the surface of protein-inorganic hybrid nanoflowers, a novel protein-derived hybrid nanoflower was prepared to investigate the adsorption behavior and reveal the function of organic and inorganic parts on the surface of nanoflowers in the adsorption process in this study. Silk fibroin (SF)-derived and copper-based protein-inorganic hybrid nanoflowers of SF@Cu-NFs were prepared through self-assembly. The product was characterized and applied to adsorption of heavy metal ion of Pb(II). With Chinese peony flower-like morphology, the prepared SF@Cu-NFs showed ordered three-dimensional structure and exhibited excellent efficiency for Pb(II) removal. On one hand, the adsorption performance of SF@Cu-HNFs for Pb(II) removal was evaluated through systematical thermodynamic and adsorption kinetics investigation. The good fittings of Langmuir and pseudo-second-order models indicated the monolayer adsorption and high capacity of about 2000 mg g^−1^ of Pb(II) on SF@Cu-NFs. Meanwhile, the negative values of $\Delta_{r}G_{m\left(T\right)}^{\theta }$ and $\Delta_{r}H_{m}^{\theta }$ proved the spontaneous and exothermic process of Pb(II) adsorption. On the other hand, the adsorption mechanism of SF@Cu-HNFs for Pb(II) removal was revealed with respect to its individual organic and inorganic component. Organic SF protein was designated as responsible ‘stamen’ adsorption site for fast adsorption of Pb(II), which was originated from multiple coordinative interaction by numerous amide groups; inorganic Cu_3_(PO_4_)_2_ crystal was designated as responsible ‘petal’ adsorption site for slow adsorption of Pb(II), which was restricted from weak coordinative interaction by strong ion bond of Cu(II). With only about 10% weight content, SF protein was proven to play a key factor for SF@Cu-HNFs formation and have a significant effect on Pb(II) treatment. By fabricating SF@Cu-HNFs hybrid nanoflowers derived from SF protein, this work not only successfully provides insights on its adsorption performance and interaction mechanism for Pb(II) removal, but also provides a new idea for the preparation of adsorption materials for heavy metal ions in environmental sewage in the future.

## 1. Introduction

With fast growing activities of urbanization and industrialization, heavy metal ion (HMI) contamination in water environments has been widely brought by the rapid economic development [1]. Due to their rapid accumulation in the food chain and non-biodegradable properties, HMIs are regarded as one of the most serious contamination sources with highly toxicity and carcinogenicity even at trace amount exposure [2]. Pb(II) is an often encountered HMI which has been widely used in industries of batteries manufacturing, shipbuilding, oil mining, etc. [3]. The large amount of Pb(II) discharge in water environment and Pb(II) accumulation in human body can lead to physical defects such as nephropathy, hepatopathy, and encephalopathy [4,5]. Even more, high concentration of lead ions will do harm to children’s health [6]. According to the guidelines set by WHO and EPA, the permissible limit of Pb(II) in portable water should not exceed 0.05 mg L^−1^ [7,8]. On account of the serious threatening on the ecosystem’s sustainable development and human health, the removal of Pb(II) from waste water has become an urgent problem and a mandatory task for environmental protection [9].

Various treatment techniques—such as chemical reduction [10], biological conversion [11], membrane separation [12,13], and adsorption treatment [14]—have been developed and applied to remove HMIs during the past decades. With obvious advantages of high efficiency, cost-effectiveness and simple operation, adsorption technology has been regarded as one of the most effective and competitive methods for HMIs treatment [15,16,17,18]. Consequent, the development of functional adsorbent material and the application to efficient HMIs removal are highly desirable for water pollution treatment.

So far, a great number of materials—including lignin [19], biochar [20], chitosan [21], fabrics [22], soil [23], metal-organic frameworks (MOFs) [24], graphene oxide (GO) [25], and nanomaterials (such as nanofiber, nanobubble, and nanotube) [26,27,28]—have been studied and prepared as adsorbents for HMIs removal. Among these materials, organic–inorganic hybrid nanoflowers (HNFs) is newly developed functional material and has received considerable attention due to its distinctive physiochemical characteristics. By binding inorganic nanoparticles to organic components, HNFs show properties of simple product synthesis and high biomolecule efficiency comparing with the pure organic nanoflowers [29] and inorganic nanoflowers [30]. Since Ge et al. [31] first reported the preparation of BSA-incorporated Cu_3_(PO_4_)_2_ nanoflowers, biomaterials-based HNFs have attracted increasing interest and many researches have focused on their biochemical applications of biosensing [32,33,34], biocatalysis [35,36,37], and drug delivery [38].

The organic component and the preparation method were two important aspects which would have great influence on the structure, morphology and property of the HNFs composites. For organic component of HNFs, protein is usually selected as a typical biological material for HNFs fabrication owing to its unique chemical structure and special biological property. A series of proteins, including serum albumin (BSA) [31], glucose oxidase (GOx), horseradish peroxidase (HRP) [39], and immunoglobulin G (Ig G) [40] have been employed to prepare HNFs. Although these nanoflowers show excellent performances, the products generally suffer from the disadvantages of high price and difficult acquisition of protein, which greatly limits the HNFs products in actual applications. 

Silk fibroin (SF), a facile and low-cost protein which is obtained from the silkworm, is a well-known and widely-used natural macromolecular protein. During the past thousands of years, SF has been considered as an excellent raw material for the traditional use in textile industries [41]. Nowadays, the attractive properties of SF protein—such as good mechanism stability [42], superior biocompatibility [43], and excellent optic performances [44]—have made SF effective use in bioelectronic substrate [45], optical sensor [46], drug delivery [47,48], and so on. Consequently, SF protein has been regarded as an excellent candidate of organic biomolecules for HNF fabrication [49]. 

Besides the organic component, the preparation method is also of great importance for the HNFs preparation. If the biomolecules are improperly bonded or immobilized with the organic component, the prepared HNFs usually exhibit lower biomolecule activity, enhanced biomolecule mass-transfer limitations, and unfavorable conformational changes in the biomolecules [50]. Compared with conventional immobilization methods (such as covalent bond [51], physical trap [52]) and new fabrication techniques (such as welding [53], nanoimprinting [54]), self-assembly process have proven to show characteristics of simple synthesis, high efficiency, and bright prospect of enhancing stability, activity, and even selectivity of biomolecules for HNFs fabrication [55].

In this work, copper–protein hybrid nanoflowers by employing SF protein as natural biomaterial and copper phosphate as inorganic component are fabricated for efficient Pb(II) treatment by self-assembly method. The prepared nanoflowers derived from SF protein, denoted as SF@Cu-HNFs thereafter, exhibit several significant advantages: (1) raw biomaterials of SF protein are easy and cheap to obtain; (2) acidic amino acids in the primary structure of SF can bind cations to drive self-assembly easily; (3) abundant functional hydroxyl and amino groups are provided by SF protein for Pb(II) adsorption. The synthesized SF-based nanoflowers were characterized and applied to HMI adsorption (Pb(II), Ni(II), and Cd(II)). Compared with the adsorption performances of Cd(II) and Ni(II), the prepared SF@Cu-HNFs exhibited excellent adsorption selectivity and significant adsorption capacity for Pb(II) removal. Subsequently, the adsorption performance of SF@Cu-HNFs was systematically evaluated for Pb(II) adsorption through thermodynamic (adsorption isotherm and adsorption capacity) and adsorption kinetics investigation. Furthermore, the interaction mechanism of SF@Cu-HNFs was successfully revealed and verified for Pb(II) adsorption with respect to its individual component of organic SF protein and inorganic Cu_3_(PO_4_)_2_ crystal.

## 2. Experimental 

### 2.1. Reagents and Materials

Silk fibroin (SF) protein was purchased from Xi’an Shennong Biotechnology Co., Ltd. (Xi’an, China). The protein has been purified and used as it received. Cadmium nitrate (Cd(NO_3_)_2_) was purchased from Aladdin Reagent Co., Ltd. (Shanghai, China). Lead nitrate (Pb(NO_3_)_2_), nickel nitrate (Ni(NO_3_)_2_), copper sulphate (CuSO_4_), sodium chloride (NaCl), potassium chloride (KCl), potassium dihydrogen phosphate (KH_2_PO_4_), dibasic sodium phosphate (Na_2_HPO_4_), sodium hydroxide (NaOH), and nitric acid (HNO_3_) were obtained from Kelong Chemical Reagent Company (Chengdu, China). All the reagents were of analytical grade and used as received. Wahaha^®^ purified water (Wahaha, Hangzhou, China) was used for the preparation of solutions and throughout the experiments.

SF solution was prepared in water and the concentration was adjusted to the value as needed by experiments. Pb(NO_3_)_2_, Cd(NO_3_)_2_, and Ni(NO_3_)_2_ stock solutions of 1 × 10^3^ mg L^−1^ were prepared in water and working solutions were prepared freshly for daily use. The pH was adjusted by using small amounts of 0.1 mol L^−1^ NaOH or 0.1 mol L^−1^ HNO_3_ solutions without significantly altering the HMIs concentration. Solution pH was monitored using a pH meter (PHS-3C, Yidian Inc., Ltd., Shanghai, China).

### 2.2. Synthesis of SF@Cu-HNFs via Self-Assembly

0.01 M phosphate buffer solution (PBS, pH = 7.4): Weigh 0.135 g of potassium dihydrogen phosphate, 0.71 g of disodium hydrogen phosphate, 4 g of sodium chloride, and 0.1 g of potassium chloride in order using an analytical balance, and add an appropriate amount purified water was stirred to dissolve it. The solution was transferred to a 500 mL volumetric flask, and the volume was adjusted with purified water. It was then transferred to a reagent bottle and refrigerate at 4 °C until use.

By utilizing SF protein as natural biomaterial and Cu_3_(PO_4_)_2_ as inorganic component, the hybrid nanoflowers of SF@Cu-HNFs were synthesized according to similar methods developed for BSA-based and laccase-based nanoflowers described with some modification [31,56]. In brief, 4 mL of PBS (pH = 7.4) containing different SF concentration was firstly added with 40 µL CuSO_4_ solution (100 mM). Then resultant mixtures were gently shaken for 5 min and followed by incubation at 25 °C with different preparation time. Finally, blue hybrid nanoflowers were collected, washed, with deionized water several times and dried by vacuum freeze-drying.

### 2.3. Characterization of SF@Cu-HNFs

The morphologies of the prepared SF@Cu-HNFs products were characterized by a scanning electron microscope (SEM, EV0 MA15, Carl Zeiss, Germany) at an acceleration voltage of 20 kV. All samples were sputter-coated with gold using an E1045 Pt-coater (Carl Zeiss, Germany) before SEM observation. Elemental analysis was conducted with an energy dispersive X-ray spectrometer (EDS) equipped in the SEM. 

The crystal structures of the nanoflower products were characterized by X-ray diffraction (XRD) analysis (X Pert PRO MPD, PANalytical, Holland). Radial scans using Cu Kα radiation source at 20 mA and 40 kV were recorded in the reflection scanning mode from 2*θ* = 10 to 80° at a scanning rate of 1° min^−1^.

The chemical structures of the nanoflower products were measured by Fourier transform infrared spectroscopy (Nicolet 6700 FTIR, Thermo Fisher Scientific Corp., USA) in the range of 400−4000 cm^−1^ with KBr pellets. The thermogravimetric analysis (TGA) of nanoflowers was measured with a thermogravimetric analyzer (STA449F3, Netzsch, Germany) in a dynamic atmosphere of dinitrogen with 20 cm^3^ min^−1^ flow rate. The TGA measurements were performed with a temperature ranging from 40 to 700 °C in an alumina crucible at a rate of 5 °C min^−1^.

To evaluate the surface adsorption and interaction process of Pb(II) on nanoflowers adsorbent, the surface characteristics of SF@Cu-HNFs were furthermore investigated with the addition of different Pb(II) concentrations. The surface zeta potential was determined by dynamic light scattering (DLS) measurements (NANO ZS, Malvern Instruments Ltd., UK) equipped with the DTS Ver. 4.10 software package.

### 2.4. Adsorption Performances of SF@Cu-NFs

Under the optimized pH condition, the adsorption performances of SF@Cu-NFs were studied by thermodynamic, adsorption kinetics, and selective experiments for Pb(II) removal. The Pb(II) concentration was detected through an atomic absorption spectrophotometer (AA-7020, Beijing East West Analysis Instrument Co., Ltd., Beijing, China) during the whole experiment.

#### 2.4.1. Adsorption Kinetics Experiment

For the method, 30 mg of SF@Cu-NFs was first dispersed into 300 mL 300 mg L^−1^ Pb(II) solution under continuous stirring. The suspension was sealed and oscillated at room temperature to ensure equilibration. Then 3 mL of the suspension sample was taken from the system for filtration at a regular interval time. The residual Pb(II) concentration in the solution was also detected by AAS measurement like for thermodynamic adsorption. The amount of Pb(II) adsorbed on SF@Cu-NFs ($Q_{t}$) was calculated by subtracting the concentration of free Pb(II) at the time of t from the initial Pb(II) concentration as
(1)$$Q_{t}=\frac{\left(C_{0}-C_{t}\right)\times V}{M}$$
where *C*_0_ and *C*_t_ (mg L^−1^) were the initial and *t* time Pb(II) concentrations in liquid-phase, *V* (L) was the taken volume of dye solution, and *M* (g) was the mass of the SF@Cu-NFs adsorbent used. The data obtained were used to draw the kinetic adsorption curves for pseudo-first-order, pseudo-second-order, and intraparticle diffusion analysis.

#### 2.4.2. Adsorption Isotherm Experiment and Adsorption Thermodynamics

The thermodynamic adsorption experiment of SF@Cu-NFs was carried out by a typical batch method. First, 7 pieces of 1 mg washed and dried SF@Cu-NFs were added into 7 pieces of 10.0 mL different concentration Pb(II) solutions (5, 20, 50, 80, 100, 400, 500 mg L^−1^) placed in the tube at 298 K, respectively. Then the suspensions were sealed and were vibrated for 2 h at room temperature to ensure the complete adsorption. After filtrating the mixture for solid–liquid separation, the residual Pb(II) concentration in the solution was detected by AAS measurement. The amount of Pb(II) adsorbed on SF@Cu-NFs ($Q_{e}$) was calculated by subtracting the concentration of Pb(II) from the initial concentration as
(2)$$Q_{e}=\frac{\left(C_{0}-C_{e}\right)\times V}{M}$$
where *C*_0_ and *C*_e_ (mg L^−1^) were the initial and equilibrium Pb(II) concentrations in liquid-phase, *V* (L) was the volume of Pb(II) solution, and *M* (g) was the mass of the SF@Cu-NFs adsorbent used. The data obtained were used to draw the adsorption isotherms for Langmuir, Freundlich, and Temkin analysis.

In order to obtain the experimental parameters of adsorption thermodynamics on the Pb(II) adsorption by SF@Cu-NFs, the adsorption capacity at three different temperatures of 298 K, 308 K, and 328 K were investigated with the initial Pb(II) concentration of 100 and 500 mg L^−1^, respectively.

## 3. Results and Discussion

### 3.1. Condition Optimization for SF@Cu-NFs Preparation

In order to obtain the best formation morphology of SF@Cu-HNFs product, preparation conditions of SF concentration and reaction time were systematically investigated and optimized.

#### 3.1.1. Effect of SF Protein on SF@Cu-NFs Formation

In order to investigate the effect of SF protein on the nanoflower formation, products with and without SF were firstly prepared and characterized. For the preparation of SF@Cu-NFs in this work, SF protein was simply added into the solution containing copper ions and phosphate [57]. Meanwhile, Cu_3_(PO_4_)_2_ particles were also prepared by the similar procedures without the addition of SF solution. With 24 h preparation time, the photography and SEM images were shown in Figure 1a for SF@Cu-NFs products prepared with 400 mg L^−1^ SF and were shown Figure 1b for Cu_3_(PO_4_)_2_ particles without SF, respectively. As it can be seen, although the appearances were similar (insets in Figure 1a,c), the microscopic images showed significant differences between SF@Cu-NFs products and Cu_3_(PO_4_)_2_ particles. As shown in Figure 1a,b, SF@Cu-NFs products were successfully formed by self-assembly with uniform flower-like morphology similar to the Chinese national flowers peony (Inset in Figure 1b). However, Cu_3_(PO_4_)_2_ particles with irregular flakes and uneven particle size distribution were formed without SF addition (as shown in Figure 1c,d). The Cu_3_(PO_4_)_2_ particles were not flower-like but Chinese tremella like with loose structures (Inset in Figure 1d). The above results confirmed that SF protein was a key factor and had beneficial effects for the flower-like nanoflowers formation. 

#### 3.1.2. Effect of SF Concentration on SF@Cu-NFs Formation

Under the condition of 12 h reaction time, the effect of SF concentration on the product formation was investigated ranging from 50 to 400 mg L^−1^. As shown in Figure 2, it was notable to observe that there was a great variation in morphology of SF@Cu-NFs by regulating the protein concentrations. Interestingly, the SF@Cu-NFs became smaller with the increase of SF concentration (Figure 2a,b,e,f), which may be caused by the increasing number of nucleation sites on the SF molecular. However, some SF@Cu-NFs products would bind with each other with SF concentration at 200 and 400 mg L^−1^ (Figure 2c,d,g,h), which was also disadvantageous for the adsorption with decreasing the SF@Cu-NFs number and surface. As a result, 100 mg L^−1^ SF concentration was selected to prepare the SF@Cu-NFs with the proper product size and superior shape.

#### 3.1.3. Effect of Reaction Time on SF@Cu-NFs Formation

In order to study the formation process of the three-dimensional hierarchical structures, the effect of preparation time on the product formation was investigated with the addition of 100 mg L^−1^ SF concentration. Experiments were carried out by collecting samples from the reaction mixture and observing intermediates and products at different time intervals. As shown in Appendix A, the PBS solution containing SF changed into blue after the addition of CuSO_4_. Then the solution became turbid blue after 10 min, indicating the production of NFs products. The SEM images in Figure 3a–d and insets showed the appearance and solution changes of SF@Cu-NFs product ranging from 30 min to 24 h. The diameter distribution of products from 0 to 24 h was shown in Appendix A. The process of product formation can be divided into four corresponding stages. 

The first stage is initial stage with reaction time from 10 to 30 min. At this stage, blue fine and visible particles began to appear in the solution (shown as inset in Figure 3a). As observing from corresponding SEM shown in Figure 3a, primary crystal of Cu_3_(PO_4_)_2_ was formed (encircled in red circle) and SF protein molecules complexed with Cu^2+^ on its surface (encircled in green circle). The product composited mainly through the coordination of amide groups in the protein backbone and was beneficial for the formation of larger nanosheet petals. 

The second stage is growth stage with reaction time from 30 min to 6 h. At this stage, the blue particles grew bigger and blue flocculent precipitation was observed in the solution (shown as inset in Figure 3b). As observing from corresponding SEM shown in Figure 3b, a series of SF@Cu-NFs products with complete flower-like shape have been formed (encircled in red circle). However, the morphology SF@Cu-NFs products were not uniformed and there were still some small petal products (encircled in green circle). This result indicated the nanoflowers needed to grow further.

The third stage is the formation stage with reaction time from 6 h to 12h. At this stage, small product particles furthermore grew up and deposited to the bottom ((shown as inset in Figure 3c)). As observing from corresponding SEM shown in Figure 3c, SF@Cu-NFs products with uniform flower-like shape and size were formed. 

The fourth stage is the overgrowth stage with reaction time from 12 h to 24 h. At this stage, some SF@Cu-NFs grew up and deposited to the bottom (shown as inset in Figure 3d). This was because polar side chain of SF could promote the formation of large folding nanosheets through its hydroxyl, carboxyl, and amino groups. As observing from corresponding SEM shown in Figure 3d, the further growth made some SF@Cu-NFs bind with each other to form much bigger product (encircled in green circle). The overgrowth effect decreased the SF@Cu-NFs number and surface, which was disadvantageous for the adsorption. As a result, the incubation time of 12 h was considered to be the optimum preparation time for the SF@Cu-NFs products. The schematic illustration of the formation process of SF@Cu-NFs products at preparation time of 12 h was shown in Figure 3e.

### 3.2. Characterization of SF@Cu-NFs Product

#### 3.2.1. Surface Morphology Measurement by LSCM and SEM Image

The morphologies of the synthesized SF@Cu-NFs were observed using and SEM images (shown in Figure 4a,b). As seen from SEM images, the prepared SF@Cu-NFs displayed highly peony flower-like morphology with diameters of about 50 µm.

Meanwhile, the EDS result was shown in Figure 4c and the related EDS data were listed in Appendix A. The results identified the chemical species of the SF@Cu-NFs products and confirmed the presence of Cu, P, C, and O. The C and O can be attributed to SF protein and Cu, P and O can be attributed to Cu_3_(PO_4_)_2_. Meanwhile, the appearance of Cl and Na may be brought by the residual PBS. However, the N element was not detected in the prepared SF@Cu-NFs product by EDS analysis. Because SF protein was generally considered as nitrogen rich [58], the abnormal absence of N may be attributed to the abundant O and C in the SF@Cu-NFs product, which would cover up the N peak in EDS.

#### 3.2.2. Chemical Structure Investigation by FTIR and XRD

The phase structures of the as-prepared Cu_3_(PO_4_)_2_ and SF@Cu-NFs were investigated by the XRD analysis and shown in Figure 5a. As observed, the diffraction peaks of SF@Cu-NFs and unmodified Cu_3_(PO_4_)_2_ were in good agreement with the Joint Committee on Powder Diffraction File data for Cu_3_(PO_4_)_2_ (File NO. 00-022-0548). As a result, it could be concluded that the petals in the hybrid nanoflowers were formed by regular arrangement of Cu_3_(PO_4_)_2_ crystals and the inorganic composition of SF@Cu-NFs was Cu_3_(PO_4_)_2_.

By assigning peaks to various groups and bonds, the detail FTIR spectra of Cu_3_(PO_4_)_2_ (spectrum i), SF@Cu-NFs product (spectrum ii) and SF protein (spectrum iii) were shown in Figure 5b. As it can be seen, SF@Cu-NFs showed (1) weak peaks at 563 cm^−1^ and 603 cm^−1^ corresponding to flexural vibration of P-O, (2) strong peaks at1035 cm^−1^ corresponding to stretching vibration of P-O [59,60]. These signals corresponded to the asymmetric and symmetric stretching vibrations of PO_4_^3−^, which were also present in Cu_3_(PO_4_)_2_ (spectrum i). Meanwhile, SF@Cu-NFs also showed (1) peak at 1638 cm^−1^ corresponding to stretching vibration of C-O originated from amide Ⅰ, (2) peak at 1536 cm^−1^ corresponding to superposition of bending vibration of N-H and stretching vibration of C-N originated from amide Ⅱ, (3) peak at 1421 cm^−1^ corresponding to bending vibration of N-H originated from amide Ⅲ [61]. These signals corresponded to the major amide bands originating from SF protein, which were also present in SF protein (spectrum iii). Compared with Cu_3_(PO_4_)_2_ and SF, SF@Cu-NFs showed no significant changes before and after the formation. Without new absorption peaks or obvious peak shifts, the above results indicated that SF protein was fixed by self-assembly rather than by covalent bonds. Meanwhile, the structural integrity of the silk fibroin protein remained intact after product formation, which furthermore identified the successful preparation of SF@Cu-NFs.

#### 3.2.3. Component Analysis by TGA

In order to clarify the developed nanocomposites construction of the prepared product, TGA was used to demonstrate the existence of SF protein in SF@Cu-NFs based on the gravity measurement, shown as spectrum (i), (ii), and (iii) in Figure 6 and Table 1 for Cu_3_(PO_4_)_2_, SF@Cu-NFs and SF protein, respectively. As can be see, although both the weight loss of Cu_3_(PO_4_)_2_ and SF@Cu-NFs could be divided into three stages, but there were some differences between them. For Cu_3_(PO_4_)_2_ as shown in spectrum (i), its weight loss included (1) due to the physical combination water; (2) due to part of chemical crystalline water; (3) due to rest of chemical crystalline water. However, the three weight loss stages of SF@Cu-NFs as shown in spectrum (ii) included (1) due to the physical combination water; (2) due to the chemical crystalline water; (3) due to the decomposition of amino acid residues and major peptide chains of SF protein. The weight loss of SF protein showed three stages, including (1) due to the physical combination water; (2) due to chemical crystalline water; (3) due to decomposition of amino acid residues and major peptide chains. The weight loss of chemical crystalline water showed the difference between Cu_3_(PO_4_)_2_ and SF@Cu-NFs, which was mainly resulted from the different water content. This was mainly because Cu_3_(PO_4_)_2_ was well-known to contain more crystalline waters like CuSO_4_·5H_2_O crystal but protein@Cu-NFs were generally considered to contain three crystalline waters in the form of Cu_3_(PO_4_)_2_·3H_2_O [31,39].

### 3.3. SF@Cu-NFs Adsorption Performance for Pb(II)

#### 3.3.1. Investigation of pH Influence on Pb(II) Adsorption

The pH value is a very key factor for the investigation of HMIs adsorption, and the pH effect on the adsorption of Pb(II) by the prepared SF@Cu-NFs was investigated from the following two aspects in this work.

On one hand, pH can affect the existing form of the adsorbate metal ion. Generally speaking, HMI would produce hydrolysis with the change of pH and Pb(II) would exit with different forms at different pH (free ionic Pb^2+^ or solid Pb(OH)_2_). In order to precisely study the adsorption process, Pb(II) should be excellently prevented from precipitating to be Pb(OH)_2_). Consequently, the initial pH must be controlled for the accurate monitoring of Pb(II) adsorption. The pH can be calculated for Pb(II) by the following precipitation–dissolution equilibrium
(3)$$\mathrm{Pb}{\left(\mathrm{OH}\right)}_{2}↔{\mathrm{Pb}}^{2+}+2{\mathrm{OH}}^{-}$$

Its equilibrium constant was
(4)$$K_{sp\left(\mathrm{Pb}{\left(\mathrm{OH}\right)}_{2}\right)}=\left[{\mathrm{Pb}}^{2+}\right]{\left[{\mathrm{OH}}^{-}\right]}^{2}$$

Then, pOH and pH were calculated to be
(5)$$p\mathrm{OH}=-lg\sqrt{\frac{K_{sp\left(\mathrm{Pb}{\left(\mathrm{OH}\right)}_{2}\right)}}{\left[{\mathrm{Pb}}^{2+}\right]}}$$
and
(6)$$pH=14+lg\sqrt{\frac{K_{sp\left(\mathrm{Pb}{\left(\mathrm{OH}\right)}_{2}\right)}}{\left[{\mathrm{Pb}}^{2+}\right]}}$$
with $K_{sp\left(\mathrm{Pb}{\left(\mathrm{OH}\right)}_{2}\right)}=1.2\times 10^{-15}$ and trace amount $\left[{\mathrm{Pb}}^{2+}\right]<10^{-7}\;\mathrm{mol}\cdot L^{-1},$ the pH for Pb(II) adsorption should be pH<10 by Equation (6).

On the other hand, pH can affect the surface charge of the adsorbent material, which is a main factor influencing the adsorption capacity [62]. Then the zeta potentials of SF@Cu-NFs were analyzed at different pH conditions by calculating the average of 10 measurements. The test shown in Appendix A indicates that the surface of the prepared SF@Cu-NFs adsorbent was positively charged at pH < 5 but negatively charged at pH > 5. The isoelectric point of SF@Cu-NFs was calculated to be pH = 4.2, at which the dispersion system of SF@Cu-NFs showed the lowest stability. Meanwhile, SF@Cu-NFs showed the highest stability with the highest absolute zeta potential of 12.83 mV at pH = 5.0, which was expected to have the best adsorption result.

Combinding the results of above two aspects, the influences of pH value on the adsorption were correspondingly investigated in the range of pH = 4.0–9.0 for 200 mg L^−1^ Pb(II) and the responses of adsorption capacity were shown in Appendix A. It was noted that the adsorption capacities increased sharply with increasing pH value from 4.0 to 5.0, and then decreased slowly from 5.0 to to 9.0. As a result, the prepared SF@Cu-NFs showed the maximum adsorption capacity at pH = 5.0, which was consistent with above result of zeta potential investigation. Subsequently, pH = 5.0 was selected as the optimum condition to obtain the best Pb(II) adsorption.

#### 3.3.2. Evaluation of Adsorption Kinetics 

The kinetics investigation is important to choose the optimal operating condition on practical systems for HMIs removal [63]. To get a deeper understanding of the adsorption process of Pb(II) on SF@Cu-NFs, three typical kinetic models including pseudo-first-order (Equation (7)), pseudo-second-order (Equation (8)) and intraparticle diffusion (Equation (9)) models were used to analyze the experimental data
(7)$$\ln\left(Q_{e}-Q_{t}\right)=\ln Q_{e}-k_{1}t$$
(8)$$\frac{t}{Q_{t}}=\frac{1}{k_{2}Q_{e}^{2}}+\frac{t}{Q_{e}}$$
(9)$$Q_{t}=C+k_{n}t^{0.5}$$
where *t* (min) is the adsorption time; $Q_{e}$ and $Q_{t}$ (mg g^−1^) are the Pb(II) amount adsorbed at equilibrium; and *k*_1_ (min^−1^), *k*_2_ (g mg^−1^ min^−1^), and *k*_n_ (mg g^−1^ min^−1/2^) are the rate constants of pseudo-first-order, pseudo-second-order, and intraparticle kinetics models, respectively. The adsorption capacity at different time was indicated in Figure 7a and the kinetic experimental data investigated by the three kinetic models were shown in Figure 7b–d, respectively. The fitting equations and kinetic parameters for the adsorption of Pb(II) by the prepared SF@Cu-NFs, as calculated from the plots of above three models, were listed in Table 1.

Pseudo-first-order (shown in Figure 7b) and pseudo-second-order (shown in Figure 7c) models were generally used to predict equilibrium adsorption capacity. With high correlation coefficient values of $R_{2}^{2}=$0.99 > $R_{1}^{2}=$0.98, the results indicated both the pseudo-second-order model provided a better fitting effect on the experimental data, demonstrating the calculated value of the pseudo-second-order model was closer to the actual value than that of pseudo-first-order model. As a result, the adsorption capacity of 300 mg L^−1^ Pb(II) was then estimated to be 2528.36 mg g^−1^ by the prepared SF@Cu-NFs. As shown in Table 2, the relative error between the calculated capacity of 2528.36 mg g^−1^ and experiment result of 2407.00 mg g^−1^ was evaluated to be 4.8%, which indicating the good agreement for them.

The intraparticle diffusion model was usually employed to examine the controlling mechanism such as transfer and chemical reaction for the adsorption process. As shown in Figure 7d, the fitting curve of intraparticle diffusion can be divided into two linear parts, indicating that the adsorption process consists of two steps. The first stage belongs to boundary layer adsorption, which is the diffusion of Pb(II) adsorbate from solution to SF@Cu-NFs surface. In the second stage, Pb(II) ion passes through the boundary layer to further react inside the SF@Cu-NFs adsorbent, which belongs to the intraparticle diffusion. Because the straight lines of the two stages do not pass through the origin of coordinate axis, it shows that the adsorption process is controlled by both intraparticle diffusion and boundary layer diffusion. 

#### 3.3.3. Adsorption Isotherm Experiment and Adsorption Thermodynamics

At the above optimal condition of pH = 5, the adsorption thermodynamics were further investigated by changing the initial Pb(II) concentrations with 1 × 10^−3^ g mL^−1^ SF@Cu-NFs adsorbent. The adsorption performances of Pb(II) on SF@Cu-NFs were studied by the following Langmuir (Equation (10)), Freundlich (Equation (11)) and Temkin (Equation (12)) models.
(10)$$\frac{C_{e}}{Q_{e}}=\frac{C_{e}}{Q_{\max}}+\frac{1}{K_{L}Q_{\max}}$$
(11)$$\ln Q_{e}=\ln K_{F}+\frac{1}{n}\ln C_{e}$$
(12)$$Q_{e}=A\ln C_{e}+B$$
where *K*_L_, *K*_F_, and A are the equilibrium constants of Langmuir, Freundlich and Temkin adsorption, respectively; *C*_e_ is the equilibrium concentration of Pb(II); *Q*_e_ and *Q*_max_ are the amount of equilibrium adsorption capacity and the maximum adsorption capacity of Pb(II), respectively; The value of *n* > 1 suggests a normal Langmuir isotherm and *n* < 1 suggests the cooperative adsorption, respectively [64].

The adsorption of Pb(II) was investigated with different initial concentrations of 5–500 mg L^−1^ at different temperatures of 298 K. Meanwhile, due to the high adsorption ability of SF@Cu-NFs, almost all the Pb(II) in the solution were adsorbed completely in the low concentration of 5–50 mg L^−1^. The free concentration of Pb(II) in the final solution could not be effectively detected and the equilibrium concentration of Pb(II) was thereafter regarded to be 0 during this concentration stage. As a result, the Langmuir, Freundlich, and Temkin adsorption isotherms of Pb(II) adsorption were efficiently fitted in high Pb(II) concentration of 80–500 mg L^−1^ and shown in Appendix A, respectively.

For clarity, the fitting equations and parameters of Langmuir model, Freundlich model and Temkin model for Pb(II) by SF@Cu-NFs were summarized and listed in Table 3. It is found that Langmuir model provided better fitting to the equilibrium data than that of Freundlich and Temkin models with a higher correlation coefficient of 0.98, indicating that the adsorption of Pb(II) on the prepared SF@Cu-NFs belonged to monolayer adsorption instead of multilayer adsorption. Since the Langmuir model suggested that molecules are adsorbed uniformly, it can be deduced that the prepared SF@Cu-NFs were fairly homogeneous with SF protein assembly.

In order to evaluate the treat ability of the prepared SF@Cu-NFs for Pb(II) adsorption, the maximum adsorption capacity in this work was compared the results obtained by some other adsorbents which were reported previously. The results were listed in Appendix A. As compared, SF@Cu-NFs indicated as an excellent adsorbent for Pb(II) treatment with the $Q_{max}$ as high as 1908 mg g^−1^, which was about 3–20 folds than that of the other adsorbents. As a result, SF@Cu-NFs was suggested to be a candidate for Pb(II) removal in wastewater with much higher adsorption performance. 

In order to obtain the experimental parameters of adsorption thermodynamics on the Pb(II) adsorption by SF@Cu-NFs, the adsorption capacity and equilibrium constant at different temperatures were shown in Figure 8a,b, respectively. Then the thermodynamic data were calculated assuming the temperature-constant entropy and enthalpy of adsorption and according to the following temperature-related equations of Equation (13) to Equation (15).
(13)$$\Delta_{r}G_{m\left(T\right)}^{\theta }=-RT\ln K_{T}^{\theta }$$
(14)$$\Delta_{r}G_{m\left(T\right)}^{\theta }=\Delta_{r}H_{m}^{\theta }-T\Delta_{r}S_{m}^{\theta }$$
(15)$$K_{T}^{\theta }=\frac{Q_{e}}{C_{e}}$$

The results of $\Delta_{r}G_{m\left(T\right)}^{\theta }$, $\Delta_{r}H_{m}^{\theta }$ and $\Delta_{r}S_{m}^{\theta }$ were listed in Appendix A and several conclusions could be obtained.

First, with $\Delta_{r}G_{m\left(T\right)}^{\theta }<0$ for both 100 and 500 mg L^−1^ Pb(II), the adsorption process of Pb(II) onto SF@Cu-NFs was indicated to be spontaneous. However, the absolute value of $\Delta_{r}G_{m\left(T\right)}^{\theta }$ for Pb(II) adsorption was noted to decrease from 9.07 kJ mol^−1^ to 3.63 kJ mol^−1^ with the increase in temperature from 298 K to 328 K, which indicating that lower temperature was favored for the removal of Pb(II) by SF@Cu-NFs.

Second, with $\Delta_{r}H_{m}^{\theta }<0$ for both 100 and 500 mg L^−1^ Pb(II), the adsorption process of Pb(II) onto SF@Cu-NFs was indicated to be exothermic. However, the absolute value of $\Delta_{r}H_{m}^{\theta }$ for Pb(II) adsorption was noted to decrease from 64.30 kJ mol^−1^ to 17.91 kJ mol^−1^ with the increase in Pb(II) concentration from 100 mg L^−1^ to 500 mg L^−1^, which indicating that higher concentration would result in a mutual repulsion between mutual Pb(II).

Third, with the absolute value of $\Delta G<40\;{\mathrm{kJ}\;\mathrm{mol}}^{-1}$ at different Pb(II) concentrations and different temperatures, the observations on the adsorption of Pb(II) by the SF@Cu-NFs in present study was an obvious physical adsorption process.

#### 3.3.4. Investigation of Adsorption Selectivity for Pb(II)

Selectivity is one of the primary criteria for good adsorbents for the removal of trace amounts of heavy metals in the presence of other competing metal ions. In this work, the adsorption selectivity of SF@Cu-NFs was studied for three HMIs of Pb(II), Cd(II), and Ni(II) under the condition of 20 mL 100 mg L^−1^ HMIs with 3 mg SF@Cu-NFs adsorbent. The removal efficiency for different HMIs at different concentration were shown in Appendix A. For the three HMIs, all the adsorption processes were indicated to be rapid within the first 5 min and thereafter relatively slower by achieving the equilibrium in 90 min. The heavy metal ions adsorption efficiency (*AE*) can be calculated as
(16)$$AE\left(\%\right)=\frac{\left(C_{0}-C_{e}\right)}{C_{0}}\times 100\%$$

The adsorption efficiencies of Cd(II), Ni(II) and Pb(II) were calculated to be 23.77%, 18.76%, and 99.75%, indicating the much higher adsorption performance for Pb(II) by the prepared nanoflower. The selective factor (sf) was defined to evaluate the adsorbent selectivity as
(17)$$sf=\frac{AE_{a}}{AE_{b}}$$
where $AE_{a}$ and $AE_{b}$ were adsorption efficiencies for the superior and inferior adsorption HMIs, respectively. For the prepared SF@Cu-NFs, its selective factors of Pb(II) were calculated to be 4.2 relative to Cd(II) and 5.3 relative to Ni(II), which proved the excellent adsorption selectivity for Pb(II) by SF@Cu-NFs.

### 3.4. SF@Cu-NFs Adsorption Mechanism for Pb(II)

#### 3.4.1. Verification of Pb(II) Adsorption by SF@Cu-NFs

In order to access the interactions between SF@Cu-NFs adsorbent and Pb(II) ion, SF@Cu-NFs after Pb(II) adsorption was furthermore investigated by zeta potential, FTIR, and XRD measurements, which were respectively shown in Figure 9a,b and Appendix A.

The average zeta potentials of SF@Cu-NFs adsorbent were analyzed through DLS measurements. With the addition of different Pb(II) concentrations, the surface zeta potential measurement of SF@Cu-NFs was presented in Figure 9a. At the optimum pH = 5, the surface of the SF@Cu-NFs adsorbent was negatively charged and had an average zeta potential of about −12 mV without Pb(II). With the addition of Pb(II), an increase in the zeta potential was produced, which showed fast in lower Pb(II) concentration (indicated as the blue area) and thereafter varies rather slowly in higher Pb(II)concentration (indicated as the green area). This is a typical two-site adsorption behavior corresponding to two-type interaction dominance [65]. 

The pattern of HMI adsorption onto solid adsorbents can be attributable to the groups and bonds present on the material surface. In order to elucidate the active interaction site, FTIR spectrophotometry was performed to investigate the changes of functional groups of SF@Cu-NFs adsorbent before and after Pb(II) adsorption, which were shown as spectrum (i) and spectrum (ii) in Figure 9b, respectively. As it can be seen, the two FITR domains of SF@Cu-NFs after Pb(II) adsorption showed different groups and bonds, including (1) decreasing peaks at 1638, 1536, and 1421 cm^–1^ at SF domain, indicating the interaction between Pb(II) and functional N–H, C–O, and C–N groups; (2) increasing peaks at 1035, 603, and 563 cm^–1^ at Cu domain, indicating the interaction between Pb(II) and P-O groups. Because there was no new functional group appearing in the SF@Cu-NFs adsorbent after Pb(II) adsorption, it can be determined that the interaction between Pb(II) and SF@Cu-NFs belongs to physical but not chemical adsorption. The two bands around 2800 cm^−1^ are corresponding to stretching vibration of saturated C-H. Generally, the group with high electronegativity has strong ability of electron absorption. When it is connected with the number of carbon atoms on the carbonyl group of alkyl ketone, the electron cloud will shift from oxygen atom to the middle of double bond due to the induction effect. These increase the force constant of C=O bond, increases the vibration frequency of C=O, and shifts the absorption peak to a higher wave number. This result is also consistent to the adsorption energy obtained in thermodynamic investigation, which was calculated to be ΔG < 40 kJ mol^−1^.

As presented in Appendix A, XRD patterns of SF@Cu-NFs were measured before and after Pb(II) uptake. Compared to the diffraction peaks of hybrid nanoflowers before adsorbing Pb(II), there were new several miscellaneous diffraction peaks at 2*θ* values of 21.5, 26.2, 27.5, 30.0, which confirmed hybrid nanoflowers successfully adsorbed heavy metal ion Pb(II).

#### 3.4.2. Mechanism Analysis of Pb(II) Adsorption by SF@Cu-NFs

Based on the results of adsorption property and adsorption characterization mentioned above, the mechanism is proposed to illustrate the elimination performance for Pb(II) by the prepared SF@Cu-NFs. The mechanism diagram is schematically presented in Figure 10, in which the chemical structure of SF protein was referenced from the previous report [66] and the electronic structure of Cu_3_(PO_4_)_2_·3H_2_O was calculated by Material Studio 7.0. Blue, yellow, red, and grey spheres designate Cu, P, O, and H atoms, respectively. The adsorption of SF@Cu-NFs for Pb(II) removal was originated from two types of adsorption sites and two kinds of interaction dominance, which can be ascribed to the individual organic SF protein and inorganic Cu_3_(PO_4_)_2_ crystal. Correspondingly, two stages of fast adsorption and slow adsorption of Pb(II) by the prepared SF@Cu-NFs was revealed and described as follows. 

Fast adsorption stage of Pb(II). For this stage, the flower ‘stamen’ of organic SF protein was designated as responsible adsorption site for fast adsorption of Pb(II) (shown as upper part in Figure 10). This kind adsorption was originated from multiple coordinative interaction produced between Pb(II) and abundant N, O elements. This interaction showed strong due to the numerous amide groups provided by SF protein. Meanwhile, the fast adsorption occurred in the shorter adsorption time (shown as the first linear part by intraparticle kinetic investigation in Figure 7c) and in the lower adsorbent concentration (shown as the first increasing part by zeta potential measurement in Figure 9a).

Slow adsorption stage of Pb(II). For this stage, the flower ‘petal’ of inorganic Cu_3_(PO_4_)_2_ crystal was designated as responsible adsorption site for slow adsorption of Pb(II) (shown as lower part in Figure 10). This kind adsorption was originated from unique coordinative interaction produced between Pb(II) and O element. This interaction showed weak due to the powerful restriction from the strong ion bond by Cu(II) elements in Cu_3_(PO_4_)_2_ crystal. Meanwhile, the slow adsorption occurred in the longer adsorption time (shown as the second linear part by intraparticle kinetic investigation in Figure 7c) and in the higher adsorbent concentration (shown as the second increasing part by zeta potential measurement in Figure 9a).

## 4. Conclusions

In this work, natural material of SF protein was used for the fabrication of protein–inorganic hybrid nanoflowers through self-assembly and the three-dimensional structure was applied to efficient adsorption of HMI Pb(II).

Through adsorption isotherms and kinetics, the adsorption performance of SF@Cu-HNFs for Pb(II) removal was systematically evaluated in detail. Langmuir and pseudo-second-order models indicated the monolayer adsorption and high capacity on the SF@Cu-NFs. Meanwhile, the adsorption thermodynamics showed that the spontaneous and exothermic process. As compared, SF@Cu-NFs indicated as an excellent adsorbent for Pb(II) treatment with the $Q_{max}$ as high as 1908 mg g^−1^, which was about 3–20 folds greater than that of the other adsorbents.

By ascribing to its individual organic and inorganic component, the adsorption mechanism of SF@Cu-NFs for Pb(II) removal was discussed and revealed with two stages of fast adsorption and slow adsorption. On one hand, the flower ‘stamen’ of organic SF protein was designated as responsible adsorption site for fast adsorption of Pb(II). On the other hand, the flower ‘petal’ of inorganic Cu_3_(PO_4_)_2_ crystal was designated as responsible adsorption site for slow adsorption of Pb(II). This result clearly indicated that the silk fibroin protein-derived hybrid nanoflower could adsorb HMI Pb(II) well because of the adsorption site on the adsorbent surface.

In this work, we further understand the adsorption behavior and interaction process of HMI Pb(II) on the surface of silk fibroin derived hybrid nanoflowers. The present study has been successful in revealing the microscopic interaction process of Pb (II) adsorption that provides a new insight on understanding the adsorption mechanism. Also, based on interfacial adsorption, it is of great significance to comprehend the development of heavy metal ion removal applications. By fabricating SF@Cu-HNFs hybrid nanoflowers derived from SF protein, this work not only successfully provides insights on its adsorption performance and interaction mechanism for Pb(II) removal, but also significantly indicates its potential applications in contamination adsorption for environmental treatment.

## Figures and Tables

**Figure 1 materials-13-01241-f001:**
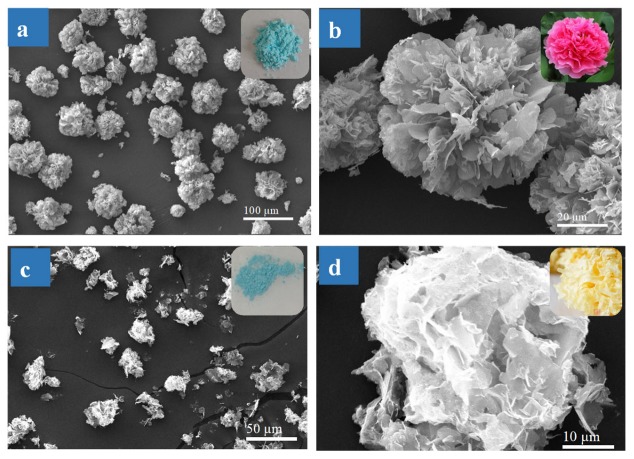
(**a**,**b**) SEM images of SF@Cu-NFs product with SF. Insets are photographies of SF@Cu-NFs and Chinese penoy flower. (**c**) and (**d**) SEM images of Cu_3_(PO_4_)_2_ particles without SF. Insets are photographies of Cu_3_(PO_4_)_2_ particles and Chinese tremella.

**Figure 2 materials-13-01241-f002:**
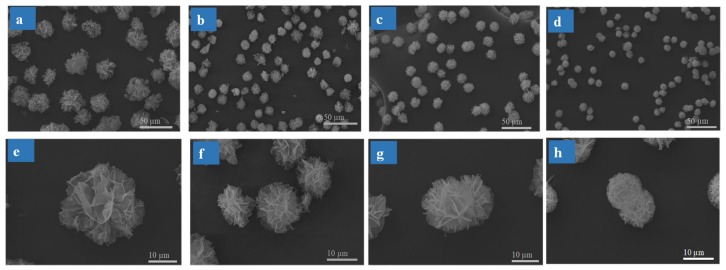
(**a**–**d**) SEM images of SF@Cu-NFs with different SF concentrations at 50 mg L^−^^1^, 100 mg L^−^^1^, 200 mg L^−^^1^ and 400 mg L^−^^1^, respectively. (**e**–**h**) The orresponding SEM images enlarged with mangnification of individual products.

**Figure 3 materials-13-01241-f003:**
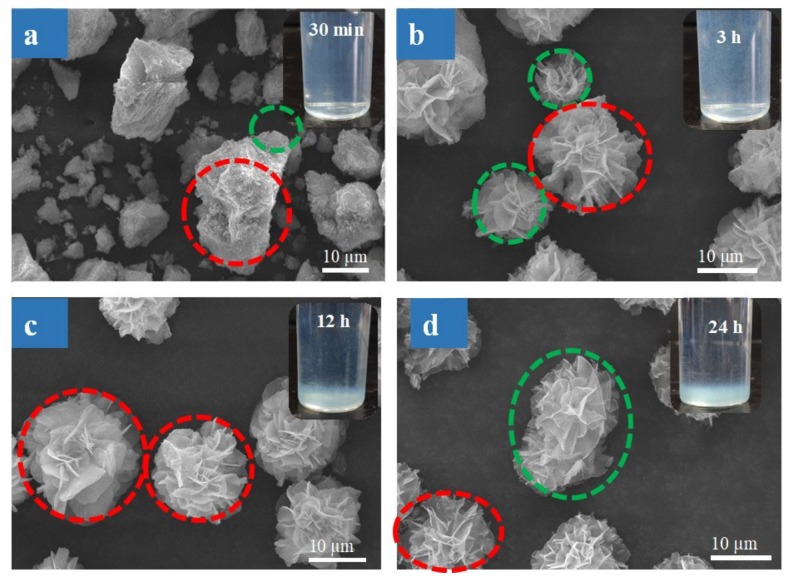
(**a**–**d**) SEM images of the nanostructures of SF@Cu-NFs products at different preparation times of 30 min, 3 h, 12 h, and 24 h. Insets are photographies of solution change. (**e**) Schematic illustration of the formation process of SF@Cu-NFs products at preparation time of 12 h.

**Figure 4 materials-13-01241-f004:**
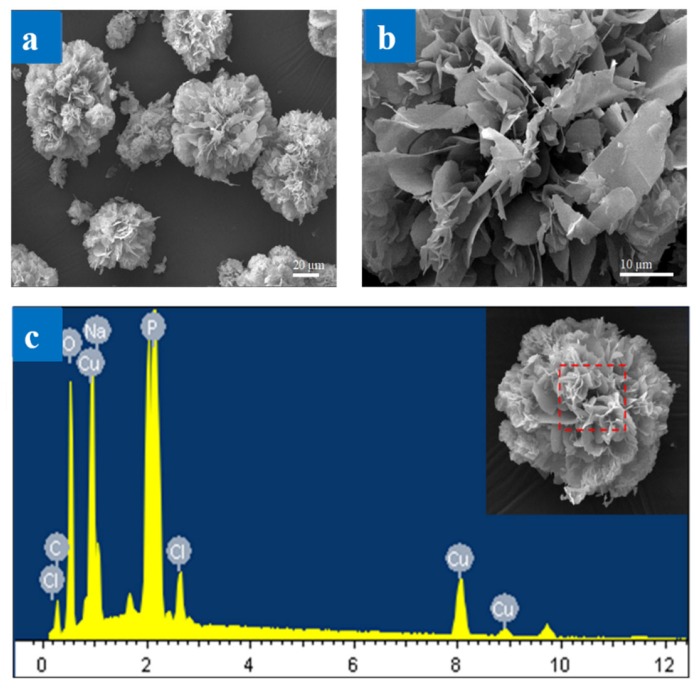
(**a**,**b**) SEM images of SF@Cu-NFs product with low and high magnification; (**c**) EDS pattern of SF@Cu-NFs product.

**Figure 5 materials-13-01241-f005:**
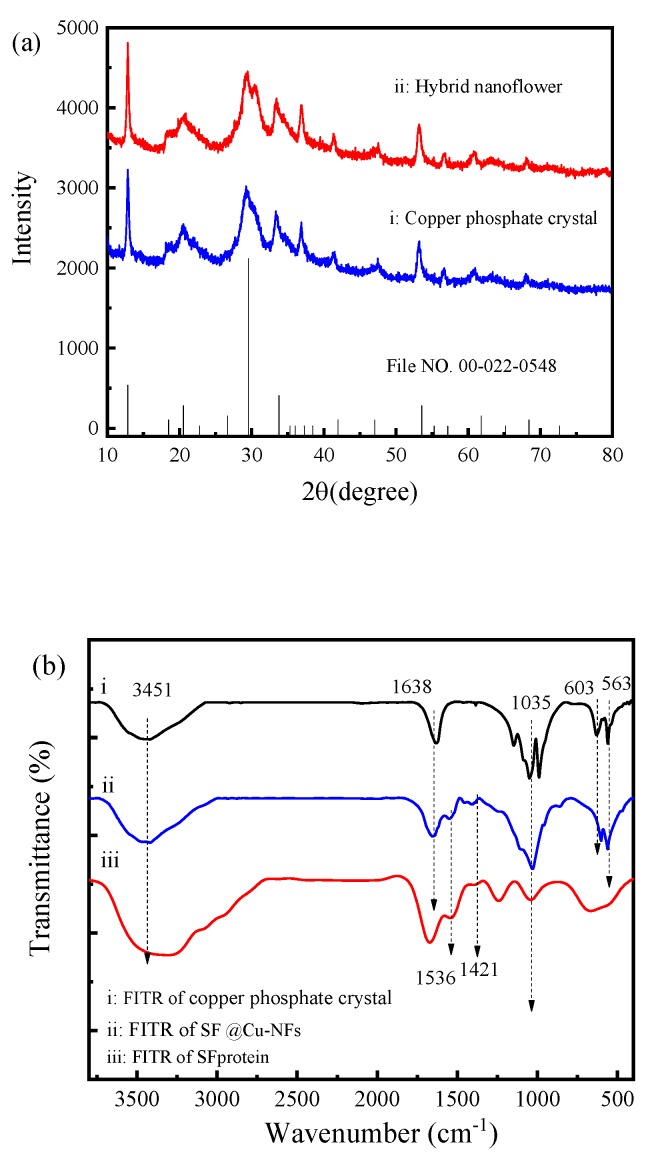
(**a**) XRD spectrum of (i) copper phosphate; (ii) SF@Cu-NFs nanoflower. (**b**) FTIR spectrum of (i) copper phosphate; (ii) SF@Cu-NFs nanoflower; (iii) SF protein.

**Figure 6 materials-13-01241-f006:**
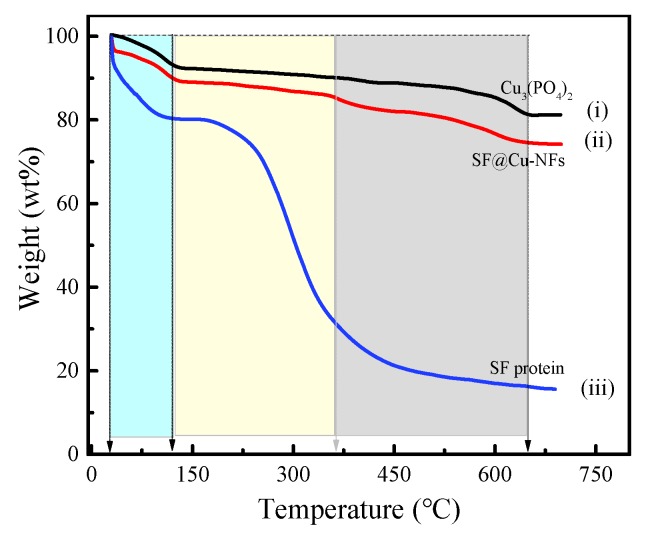
TGA spectrum of (i) Copper phosphate; (ii) SF@Cu-NFs nanoflower; (iii) SF protein.

**Figure 7 materials-13-01241-f007:**
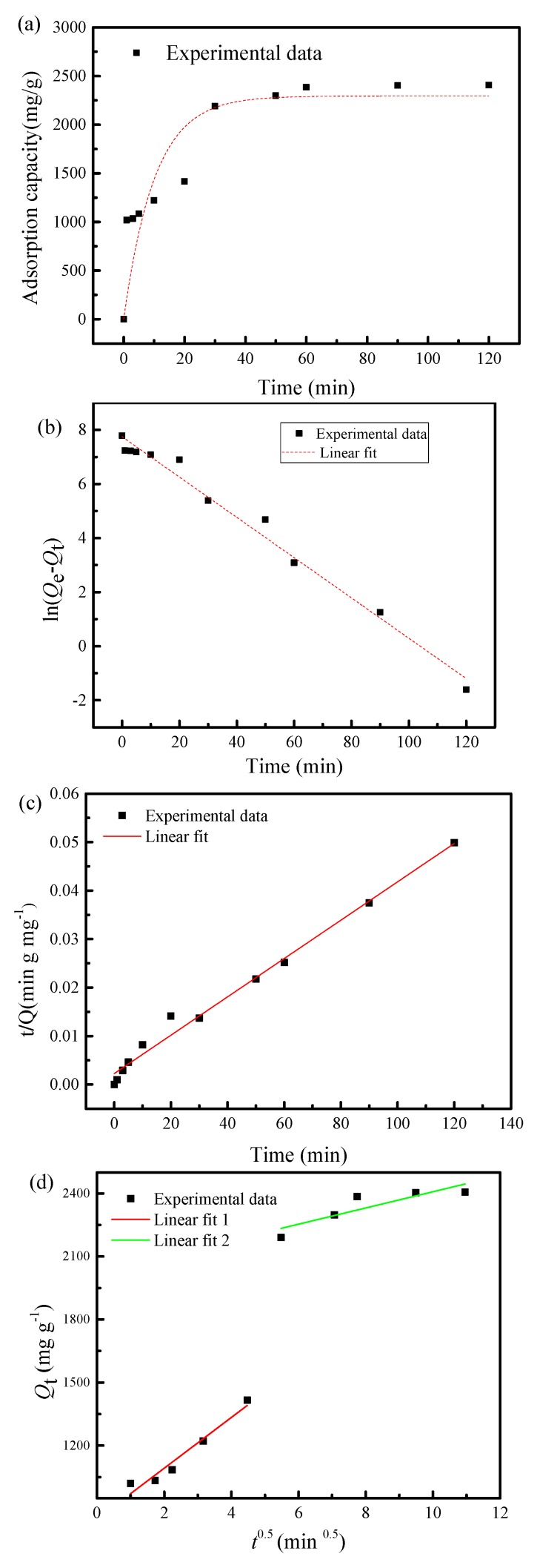
(**a**) The adsorption capacity at different times. Pb(II) removal by silk fibroin mediated nanoflowers (**b**) the pseudo-first-order; (**c**) the pseudo-second-order; (**d**) the intra-particle diffusion model sorption kinetics curves.

**Figure 8 materials-13-01241-f008:**
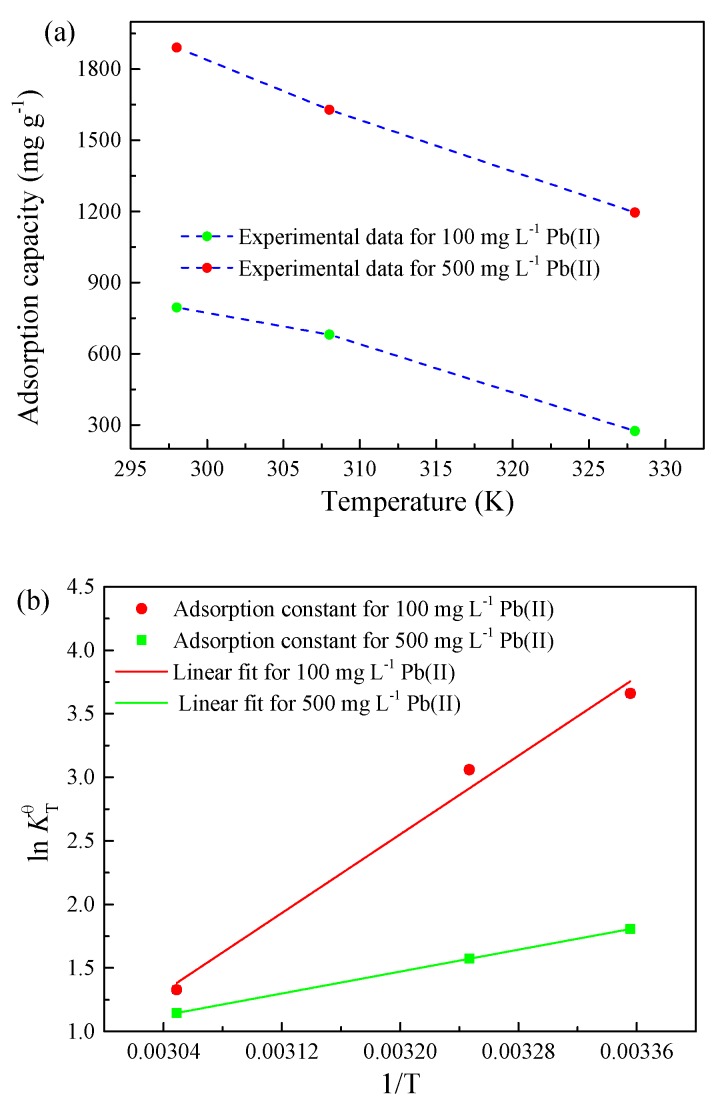
Effect of temperature on the equilibrium adsorption of Pb(II) (**a**) adsorption capacity and (**b**) adsorption constant.

**Figure 9 materials-13-01241-f009:**
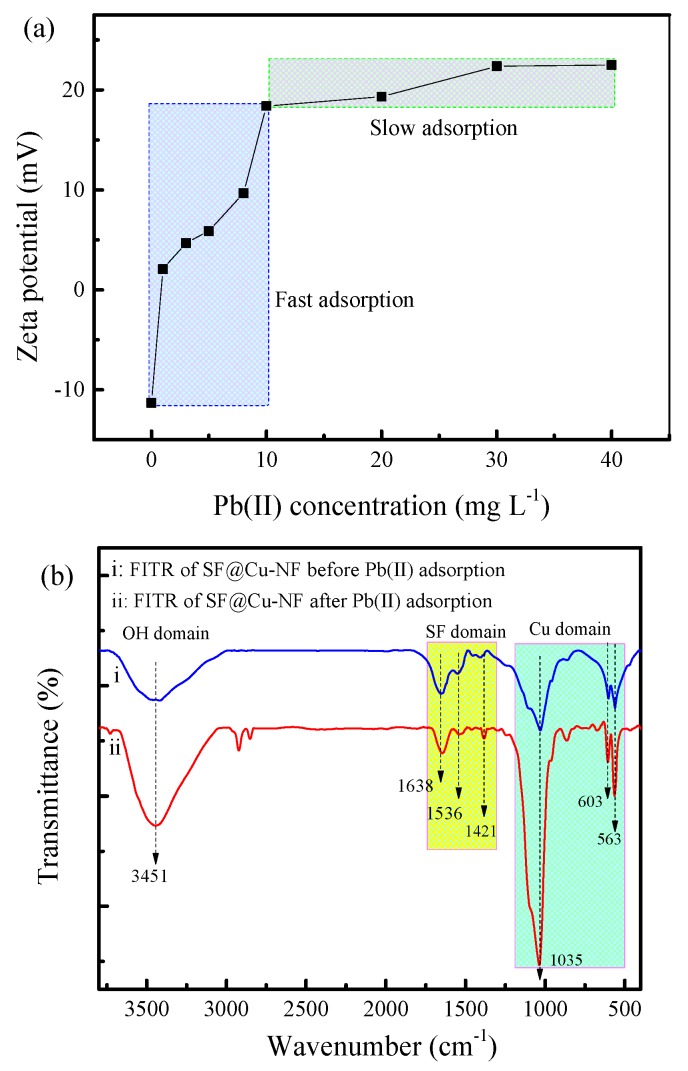
(**a**) Surface zeta potential measurement of SF@Cu-NFs at different Pb^2+^ concentrations; (**b**) FTIR spectra of SF@Cu-NFs before (spectrum i) and after (spectrum ii) Pb(II) adsorption.

**Figure 10 materials-13-01241-f010:**
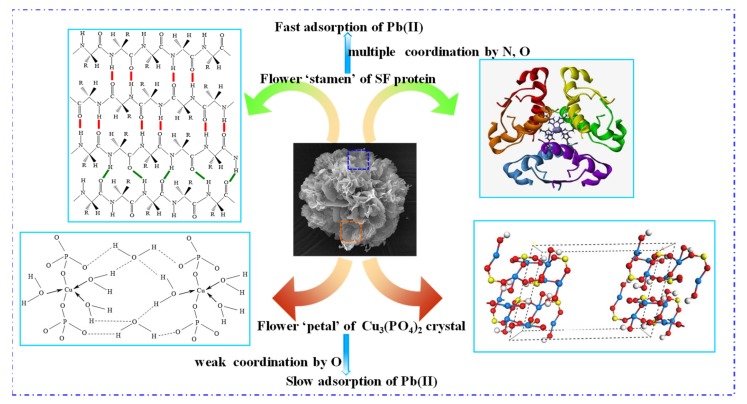
Proposed adsorption mechanism of Pb(II) by SF@Cu-NFs. Upper part indicates the fast adsorption of Pb(II) by the organic SF component and lower part indicates the slow adsorption of Pb(II) by the inorganic Cu_3_(PO_4_)_2_ component.

**Table 1 materials-13-01241-t001:** Thermogravimetric data of TGA spectrum (i) copper phosphate; (ii) SF@Cu-NFs nanoflower; (iii) SF protein.

Temperature Range	30–127 °C	127–361 °C	361–648 °C
Cu_3_(PO_4_)_2_	7.32%	2.57%	8.67%
SF@Cu-NFs	10.25%	4.30%	10.85%
SF protein	19.71%	49.63%	14.48%

**Table 2 materials-13-01241-t002:** Comparison parameters of the pseudo-first-order, the pseudo-second-order and the intra-particle diffusion models for Pb(II) adsorption by SF@Cu-NFs.

Model	Fitting Equation	Parameter	Value
Pseudo-first-order	y = −0.07x + 7.75	$Q_{e,\mathrm{cal}1\;}$(mg g^−1^)	2312.00
		*k*_1_ (min^−1^)	0.075
		$R_{1}^{2}$	0.98
Pseudo-second-order	y = 4 × 10^−4^x + 0.0023	$Q_{e,\mathrm{cal}2\;}$(mg g^−1^)	2528.36
		*k*_2_ (g mg^−1^ min^−1^)	6.92 × 10^−5^
		$R_{2}^{2}$	0.99
Intra-particle diffusion	y = 121.00x + 850.35	*C* _1_	850.35
		$k_{\mathrm{ip}1}$	121.00
		$R_{\mathrm{ip}1}^{2}$	0.94
	y = 38.61x + 2022.19	*C* _2_	2022.19
		$k_{\mathrm{ip}2}$	38.61
		$R_{\mathrm{ip}2}^{2}$	0.70
Result in this work		$Q_{e,\exp}$(mg g^−1^)	2407.00

**Table 3 materials-13-01241-t003:** Comparison parameters of Langmuir, Freundlich and Temkin models for Pb(II) adsorption by SF@Cu-NFs.

Fitting Model	Langmuir Model y = 5.24 × 10^−4^x + 0.11			Freundlich Model y = 0.1895x + 6.36			Temkin Model y = 230.24x + 409.93		
Parameter	$Q_{max}$ (mg g^−1^)	*K*_L_ (mg L^−1^)	$R^{2}$	*K*_F_ (mg^n−1^ g^−1^ L^−n^)	n	$R^{2}$	A	B	$R^{2}$
Value	1908.39	4.76 × 10^−3^	0.98	578.25	5.277	0.79	230.24	409.93	0.77

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
