# Peer review of "Investigation on the Adsorption-Interaction Mechanism of Pb(II) at Surface of Silk Fibroin Protein-Derived Hybrid Nanoflower Adsorbent"

_materials, 2020, doi:10.3390/ma13051241_

Round 1

Reviewer 1 Report

This manuscript by Li et al. aims to assess the removal of Pb2+ by protein derived NF adsorbent. The scope of this manuscript fits well with the readership of Materials, even if large parts of the manuscript needs to be shortened and clarified, and the structure of the manuscript should be reviewed. I recommend a correction of the language of the manuscript as currently plenty of errors can be found throughout the manuscript. I recommend to accept this manuscript after major modifications. Specific comments are given below:

L12-13 and L568-569 This sentence is almost repeated several times throughout the manuscript but is very hard to understand. Please clarify the content!

L14-15 “reveal the interaction mechanism” This sentence is unclear, how an adsorbent could reveal the interaction mechanism? Please correct.

L21 “dynamic investigation” You didn’t perform any dynamic experiment in this manuscript, you performed kinetic experiments, which is very different. Please correct throughout the manuscript.

L23 “2000 g mg-1” what an exceptional adsorbent! Please correct

L25-26 This sentence doesn’t make any sense, please revise.

L34-36 None of the performed experiment “indicates its potential applications […]” please moderate and revise as this sentence is currently hard to understand.

L40 “quick accumulation and non-biodegradable nature in food chain” please recheck the precise meaning and the word’ order, I think that there is a mistake.

L45-46 “Pb(II) contamination can cause serious problem of high Pb(II)” please revise this sentence which is not correctly formatted.

L50 Please check the used references in the introduction, especially when you speak about environmental concentration and fate of Pb(II). You extensively cite studies about materials/adsorption which is not correct.

L62-63 “distinctive physiochemical properties” what does it mean?

L78 What is a “facile” protein?

L100 you present abbreviations here as in L17, which are detailed in experimental section, it’s not correct.

L110-116 What is an “analytical grade” protein? What was its purity grade?

L127 Please clarify PBS for the audience.

L159 What is a “thermodynamic adsorption experiment”? Did you mean “isotherm” or “equilibrium”? Please explain. Also, in this section, please indicate the temperature(s) under investigation.

L159-173 It’s very weird to perform and present kinetic experiments after isotherm/equilibrium experiments. How have you determined the equilibrium time without kinetic? “sealed and vibrated for 2h at RT to ensure complete adsorption” L 163-164, but in fact you did not know?

L197-198 “inspire the self-assemble process” these words are meaningless, please revise.

Between 3.1.1 and 3.1.2. why did you use different SF concentrations in these two section (400 and 100 mg.L-1 respectively? Please explain?

L216 -249 This is a too long explication. Moreover, this is not convicing based on the results of FigS2. In the text you speak about a continuous growing of the particles, even if in FigS2 a strong decrease in the mean diameter between 3h and 12h? As a result the text is not consistent with the presented results, please revise.

L262-263 Here it’s not clear why 100 mgL was selected, why not 50mgL? Please clarify.

Section 3.2.1 I don’t understand the interest of this section, please limit the amount of figures and among the 5 displayed here, at least 2 are useless.

L288-289 “were in good agreement with…” it’s a good thing but as a reader I should have the possibility to observe this agreement. Please indicate the major peaks of Cu3PO4 in Fig5a.

In figure 5b I suggest to reverse ii and iii for clarity. Also, in the text you must provide references for the band assignments.

3.2.3. This section is too long to describe only three TG curves.

L321 the decomposition of aminoacids in the range 361-648°C for SF@Cu-NFs although for SF protein alone, this oxidation begins around 150°C? The gap is very big, please explain? I also think that on Fig6, the addition of DTG would be very informative.

L334-384 In this section you claim that this is a good thing to be very selective in the adsorption of Pb(II). However, it can be said that your adsorbent is not relevant for the adsorption of other contaminant such as Cd and Ni, which also raise hard toxicological issues? Please explain?

Section 3.3., please check the structure of this section as kinetic experiments should be presented first, then equilibrium experiments, and lastly adsorption selectivity experiments.

Equation (9) is not the Langmuir equation, maybe the linearized Langmuir equation.

In Figure 7, you have to present an adsorption isotherm, i.e. equilibrium concentration as a function of adsorbed amount. The figure 7a is not correct and should be changed to an adsorption isotherm.

In figure 7a, none saturation is observed? Why did you not continue to increase the starting concentrations?

In figure 7b-c-d, you must display all the experiments, as in Fig7a you present 7 points, and only 4 in figures 7b-c-d, this is unacceptable. Please correct, and it would change your main conclusions I think on the most appropriate model.

I’m not convinced about the discussion on the adsorption kinetic. It’s true that the pseudo-second-order equation presents the highest r² (as always), however, this model is not very consistent with a two-stage adsorption which is obvious based on the Figure 9an and assumed by the authors further in the manuscript. Please explain?

In figure 10b what are the two bands around 2800cm-1 after PbII adsorption? You don’t speak about that in the text?

In Figure 10c, to be honest XRD is the most convincing technique for this type of experiment. This figure can be moved in supplementary file as the variations are very weak.

L530-531 The chemisorption/physisorption assessment can be done with more sound experiments than FTIR. You have several results that concurred with physisorption. Please rewrite this section.

The conclusion is too long and verbose. In this section, you should highlight the added value of your adsorbent in comparison with other adsorbent and exhibit its potential for application. Yet, currently you perform an overview of your results which is not the aim of a conclusion. Please shorten and clarify.

Author Response

Response to Reviewer 1 Comments

Thank you for your letter and for the reviewers’ comments concerning our manuscript. Those comments are all valuable and very helpful for revising and improving our paper, as well as the important guiding significance to our researches. We have studied comments carefully and have made correction which we hope meet with approval. Revised portion are marked in red in the paper. The main corrections in the paper and the responds to the reviewer’s comments are as flowing:

Point 1: L12-13 and L568-569 This sentence is almost repeated several times throughout the manuscript but is very hard to understand. Please clarify the content!

Response 1: We have made correction according to the Reviewer’s comments. Line 12-13 the statements of “To understand and evaluate the operation mechanism of adsorption technology for heavy metal ions (HMIs), the adsorption performance at adsorbent surface is investigated.” were corrected as “For further understanding the adsorption mechanism of heavy metal ions on the surface of protein-inorganic hybrid nanoflowers,”. The corresponding contents were added in the revised manuscript (Line 12-13 in page 1).  L568-569 the statements of “It is novel to understand and evaluate the operation mechanism and efficiency of adsorption technology that the study of heavy metal adsorption performance at silk fibroin protein-derived hybrid nanoflower surface is investigated.” were deleted.

Point 2: L14-15 “reveal the interaction mechanism” This sentence is unclear, how an adsorbent could reveal the interaction mechanism? Please correct.

Response 2: We have made correction according to the Reviewer’s comments. Line 14-15 the statements of “reveal the interaction mechanism of Pb(II) at this adsorbent surface” were corrected as “reveal the function of organic and inorganic parts on the surface of nanoflowers in the adsorption process”. The corresponding contents were added in the revised manuscript (Line 14-15 in page 1).

Point 3: L21 “dynamic investigation” You didn’t perform any dynamic experiment in this manuscript, you performed kinetic experiments, which is very different. Please correct throughout the manuscript.

Response 3: We have made correction according to the Reviewer’s comments. Line 21 the statements of “dynamic” were corrected as “adsorption kinetics”. The corresponding contents were added in the revised manuscript (Line 21 in page 1). In other parts of the paper, we also made corresponding modifications.

Point 4: L23 “2000 g mg-1” what an exceptional adsorbent! Please correct

Response 4: The special three-dimensional layered structure of SF@Cu-HNFs provides many adsorption sites for the treatment of heavy metals in sewage, so the adsorption capacity of SF@Cu-HNFs is larger than that of gel, membrane and other adsorption materials.

Point 5: L25-26 This sentence doesn’t make any sense, please revise.

Response 5: Based on the characterization of SF@Cu-HNFs before and after adsorption, we analysed the effect of organic and inorganic components of SF@Cu-HNFs on the adsorption of lead, and thus studied the adsorption mechanism of SF@Cu-HNFs for removing lead.

Point 6: L34-36 None of the performed experiment “indicates its potential applications […]” please moderate and revise as this sentence is currently hard to understand.

Response 6: We have made correction according to the Reviewer’s comments. Line 34-36 the statements of “but also significantly indicates its potential applications in contamination adsorption for environmental treatment” were corrected as “but also provides a new idea for the preparation of adsorption materials for heavy metal ions in environmental sewage in the future”. The corresponding contents were added in the revised manuscript (Line 34-36 in page 1).

Point 7: L40 “quick accumulation and non-biodegradable nature in food chain” please recheck the precise meaning and the word’ order, I think that there is a mistake.

Response 7: We have made correction according to the Reviewer’s comments. Line 40 the statements of “quick accumulation and non-biodegradable nature in food chain” were corrected as “rapid accumulation in the food chain and non-biodegradable properties”. The corresponding contents were added in the revised manuscript (Line 40 in page 1).

Point 8: L45-46 “Pb(II) contamination can cause serious problem of high Pb(II)” please revise this sentence which is not correctly formatted.

Response 8: We have made correction according to the Reviewer’s comments.  Line 45-46 the statements of “Pb(II) contamination can cause a serious problem of high Pb(II) levels in the bloodstream of children” were corrected as “high concentration of lead ions will do harm to children's health”. The corresponding contents were added in the revised manuscript (Line 45-60 in page 1-2).

Point 9: L50 Please check the used references in the introduction, especially when you speak about environmental concentration and fate of Pb(II). You extensively cite studies about materials/adsorption which is not correct.

Response 9: It is really true as Reviewer suggested that L50 the used references in the introduction is not correct. Then according to the reviewer’s suggestion, L50 the references were corrected and the number of references in the revised manuscript was 9. The corresponding contents were added in the revised manuscript (Line 64 in page 2).

Point 10: L62-63 “distinctive physiochemical properties” what does it mean?

Response 10: L62-63 “distinctive physiochemical properties” means that it has optical effect, size effect, catalytic function, interface and surface effect, et al.

Point 11: L78 What is a “facile” protein?

Response 11: L78 a “facile” protein means that in the process of preparing SF@Cu-HNFs, the silk protein powder added is facile.

Point 12: L100 you present abbreviations here as in L17, which are detailed in experimental section, it’s not correct.

Response 12: We have made correction according to the Reviewer’s comments.  Line 17, “by techniques of SEM, EDS, LSCM, XRD, FTIR and TGA,” was deleted. The corresponding contents were deleted in the revised manuscript (Line 17 in page 1). Line 100, “by different techniques (SEM, EDS, LSCM, XRD, FTIR and TGA)” was deleted. The corresponding contents were deleted in the revised manuscript (Line 116 in page 3).

Point 13: L110-116 What is an “analytical grade” protein? What was its purity grade?

Response 13: Analytical grade protein: high content of main components, high purity, low interference impurities, suitable for chemical experiments. Its purity grade was 99%.

Point 14: L127 Please clarify PBS for the audience.

Response 14: PBS means phosphate buffer solution. PBS (pH = 7.4) was prepared as follows: Weigh 0.135 g of potassium dihydrogen phosphate, 0.71 g of disodium hydrogen phosphate, 4 g of sodium chloride, and 0.1 g of potassium chloride in order using an analytical balance, and add an appropriate amount purified water was stirred to dissolve it. The solution was transferred to a 500 mL volumetric flask, and the volume was adjusted with purified water. Transfer to a reagent bottle and refrigerate at 4 ° C until use. The corresponding contents were added in the revised manuscript (Line 141-145 in page 3).

Point 15: L159 What is a “thermodynamic adsorption experiment”? Did you mean “isotherm” or “equilibrium”? Please explain. Also, in this section, please indicate the temperature(s) under investigation.

Response 15: We have made correction according to the Reviewer’s comments. Then according to the reviewer’s suggestion, L159 the statements of “Thermodynamic adsorption experiment” were corrected as “Adsorption isotherm experiment and adsorption thermodynamics”. The corresponding contents were added in the revised manuscript (Line 197 in page 4). Adsorption isotherm experiment was performed at 298K. We have also made corresponding changes in the following experimental part.

Point 16: L159-173 It’s very weird to perform and present kinetic experiments after isotherm/equilibrium experiments. How have you determined the equilibrium time without kinetic? “sealed and vibrated for 2h at RT to ensure complete adsorption” L 163-164, but in fact you did not know?

Response 16: We have made correction according to the Reviewer’s comments. In fact, we first performed adsorption kinetics experiments to determine the adsorption equilibrium time, and then further conducted adsorption isotherm experiments and adsorption thermodynamic experiments. However, we reversed the order when we wrote the article. We have made correction according to the Reviewer’s comments. The corresponding contents were added in the revised manuscript (Line 184-213 in page 4).

Point 17: L197-198 “inspire the self-assemble process” these words are meaningless, please revise.

Response 17: According to the reviewer’s suggestion, the words of “to inspire the self-assemble process” in L195-197 were deleted in the revised manuscript (Line 257 in page 5).  

Point 18: Between 3.1.1 and 3.1.2. why did you use different SF concentrations in these two section (400 and 100 mg L-1 respectively? Please explain?

Response 18: In this part of 3.1.1, we were to study whether silk protein had an impact on the formation of nanoflowers, so we chose a moderate silk protein concentration of 400 mg L-1 to carry out the experiment. However, in the part of 3.1.2, we first studied the influence of silk protein concentration on the formation of nanoflowers, and then we got the appropriate concentration of 100 mg L-1 according to the experimental data. According to the optimal concentration, we studied the influence of synthesis time on the formation of nanoflower. In order to make the contents of the paper more clear and the experimental process more clear, we adjusted the order. The corresponding contents were in the revised manuscript (Line 270-391 in page 5-8).

Point 19: L216 -249 This is a too long explication. Moreover, this is not convicing based on the results of FigS2. In the text you speak about a continuous growing of the particles, even if in FigS2 a strong decrease in the mean diameter between 3h and 12h? As a result the text is not consistent with the presented results, please revise.

Response 19: According to Fig. S2, the particle size increased between 30min and 6h due to the formation of nanoflower petals. Between 6h and 12h, the flower petals containing silk protein further combined with copper phosphate to form compact layered nanoflowers, resulting in the decrease of particle size. Then according to the reviewer’s suggestion, L366 the statements of “3 h” were corrected as “6 h”. The corresponding contents were added in the revised manuscript (Line 366 in page 7). L372 the statements of “3 h” were corrected as “6 h”. The corresponding contents were added in the revised manuscript (Line 372 in page 7).

Point 20: L262-263 Here it’s not clear why 100 mg L-1 was selected, why not 50 mg L-1? Please clarify.

Response 20: When the concentration of silk protein solution was 100 mg L-1, the nanoflowers’ pattern was complete, the particle size was uniform and there was no excessive growth process. The adsorption sites on the surface of the nanoflowers were more than other concentrations. When the concentration of silk protein solution was 50 mg L-1, we could see from SEM that the particle size was not uniform, and some small particles did not form complete nanoflowers.

Point 21: Section 3.2.1 I don’t understand the interest of this section, please limit the amount of figures and among the 5 displayed here, at least 2 are useless.

Response 21: We have made correction according to the Reviewer’s comments. Two figures in Section 3.2.1 were deleted. The corresponding contents were deleted in the revised manuscript (Line 404 in page 8).

Point 22: L288-289 “were in good agreement with…” it’s a good thing but as a reader I should have the possibility to observe this agreement. Please indicate the major peaks of Cu3PO4 in Fig5a.

Response 22: We have made correction according to the Reviewer’s comments and we indicate the major peaks of Cu3PO4 in Fig5a. The corresponding contents were added in the revised manuscript (Line 429 in page 9).

Point 23: In figure 5b I suggest to reverse ii and iii for clarity. Also, in the text you must provide references for the band assignments.

Response 23: We have made correction according to the Reviewer’s comments and had reversed ii and iii in figure 5b. The references of the characteristic band of phosphate have been inserted in the revised manuscript and the number of references in the revised manuscript (Line 442 in page 9) were 59、60.  The references of the characteristic bands of silk protein have been inserted in the revised manuscript and the number of references in the revised manuscript (Line 448 in page 9) was 61.

Point 24: 3.2.3. This section is too long to describe only three TG curves.

Response 24: We have made correction according to the Reviewer’s comments. Then according to the reviewer’s suggestion, we added Table 1 to shorten the content. The corresponding contents were deleted in the revised manuscript (Line 467-478 in page 10).

Point 25: L321 the decomposition of amino acids in the range 361-648°C for SF@Cu-NFs although for SF protein alone, this oxidation begins around 150°C? The gap is very big, please explain? I also think that on Fig6, the addition of DTG would be very informative.

Response 25: Seen from the TG curves and Table 1, we have been restated this part of the article. The thermal decomposition of SF protein can be divided into three stages: one is the loss of physical binding water (30~127 °C), two is the loss of chemical crystal water (127~361 °C), and three is the decomposition of amino acids (361~648 °C). Compared with the curves of copper phosphate and SF@Cu-NFs, the second stage of silk protein is quite different, because silk protein has strong hydrophilicity and is greatly affected by temperature. The corresponding contents were restated in the revised manuscript (Line 476-479 in page 10).

Point 26: L334-384 In this section you claim that this is a good thing to be very selective in the adsorption of Pb(II). However, it can be said that your adsorbent is not relevant for the adsorption of other contaminant such as Cd and Ni, which also raise hard toxicological issues? Please explain?

Response 26: According to the characteristics of heavy metals, selective adsorption is to prepare corresponding adsorption materials or set certain solution conditions to separate heavy metals from wastewater. Because the non-selective adsorption is not selective for the removal of heavy metal pollutants, it is impossible to remove specific heavy metal ions for special wastewater. In many practical wastewater, one or two kinds of heavy metal ions are often used that there are mainly heavy metal pollutants. Therefore, from the perspective of environmental protection and resource recovery, the use of adsorbents for selective adsorption of heavy metal wastewater is of great significance. So the selective adsorption of Pb(II) by SF@Cu-NFs will not lead to raise hard toxicological issues.

Point 27: Section 3.3., please check the structure of this section as kinetic experiments should be presented first, then equilibrium experiments, and lastly adsorption selectivity experiments.

Response 27: We have made correction according to the Reviewer’s comments. First, we have carried out adsorption kinetics experiments. Secondly, the adsorption isotherms and thermodynamics experiments were carried out. At last, we did the selective adsorption experiment. The corresponding contents were restated in the revised manuscript (Line 484-761 in page 10-18).

Point 28: Equation (9) is not the Langmuir equation, maybe the linearized Langmuir equation.

Response 28: Equation (9) is the Langmuir equation. The equation can be rewritten into linear form as shown in Equation (9) and the corresponding references were in the respond reviewer 1[1, 2].

Point 29: In Figure 7, you have to present an adsorption isotherm, i.e. equilibrium concentration as a function of adsorbed amount. The figure 7a is not correct and should be changed to an adsorption isotherm.

Response 29: According to the reviewer’s suggestion, the adsorption isotherm curves could be shown in Figure 7(b-c), so Figure 7a was removed. The corresponding contents (Figure 7a) were deleted in the revised manuscript (in page 15). In order to shorten the expression of this part, we put the adsorption isotherms into the supplementary file (in page 15). The corresponding contents (Figure 7(b-c)) were removed in the revised manuscript.

Point 30: In figure 7a, none saturation is observed? Why did you not continue to increase the starting concentrations?

Response 30: Because of the large amount of SF@Cu-NFs adsorption, when the solution of heavy metal lead ion was in low concentration, lead ion can be completely adsorbed by SF@Cu-NFs materials. No adsorption saturation was observed when the concentration of lead ion was increased

Point 31: In figure 7b-c-d, you must display all the experiments, as in Fig7a you present 7 points, and only 4 in figures 7b-c-d, this is unacceptable. Please correct, and it would change your main conclusions I think on the most appropriate model.

Response 31: Because of the special structure of the SF@Cu-NFs, it has a large adsorption capacity of lead ions. When the initial lead ion solution was at low concentration, the nanoflower could absorb all the lead ions completely. When the adsorption equilibrium time was reached, the solution concentration was very low and difficult to measure. When the initial lead ion solution was in high concentration, the concentration of lead ion in the adsorbed solution could be determined accurately. So we chose the effective point to fit the adsorption isotherm.

Point 32: I’m not convinced about the discussion on the adsorption kinetic. It’s true that the pseudo-second-order equation presents the highest r² (as always), however, this model is not very consistent with a two-stage adsorption which is obvious based on the Figure 9an and assumed by the authors further in the manuscript. Please explain?

Response 32: The pseudo second-order model is different from the particle internal diffusion model (two-stage adsorption) in the Figure 9, because the parameters of their abscissa and ordinate are different. From Fig. 9a, we can see that the adsorption rate was faster about 30 minutes, and the rate was slow to gradually tend to equilibrium from 30 to 120 minutes, which is consistent with the two-stage adsorption in Fig. 9d.

Point 33: In figure 10b what are the two bands around 2800cm-1 after Pb(II) adsorption? You don’t speak about that in the text?

Response 33: Before adsorption of lead ions, the peak of infrared spectrum of SF@Cu-NFs was around 3000cm-1, which was the broad peak of hydroxyl group, mainly included the hydroxyl group of water molecules (above 3000cm-1) and hydrogen bond (3000-2800 cm-1). In the infrared spectrum of SF@Cu-NFs adsorbed lead ions, we could see that there were two new peaks around 2800 cm-1. The reason may be that when the nanoflower adsorbed lead ions, SF protein combined with hydrogen bond to cause the split of broad peak, and two new peaks appeared around 2800 cm-1.

Point 34: In Figure 10c, to be honest XRD is the most convincing technique for this type of experiment. This figure can be moved in supplementary file as the variations are very weak.

Response 34: We have made correction according to the Reviewer’s comments and the corresponding contents (Figure 10c)) were removed in the supplementary file (in page 7)

Point 35: L530-531 The chemisorption/physisorption assessment can be done with more sound experiments than FTIR. You have several results that concurred with physisorption. Please rewrite this section.

Response 35: We agree the Reviewer’s comments that the chemisorption/physisorption assessment can be done with more sound experiments than FTIR. However, in this article, we not only used the infrared characterization to explain the adsorption mechanism of nanoflowers, but also use the surface zeta potential characterization and adsorption thermodynamics data to verify.

Point 36: The conclusion is too long and verbose. In this section, you should highlight the added value of your adsorbent in comparison with other adsorbent and exhibit its potential for application. Yet, currently you perform an overview of your results which is not the aim of a conclusion. Please shorten and clarify.

Response 36: We have made correction according to the Reviewer’s comments and rewritten this part. Line 567-603 the statements were deleted and rewritten as following:

In this work, natural material of SF protein was used for the fabrication of protein-inorganic hybrid nanoflowers through self-assembly and the three-dimensional structure was applied to efficient adsorption of HMI Pb(II).

Through adsorption isotherms and kinetics, the adsorption performance of SF@Cu-HNFs for Pb(II) removal was systematically evaluated in detail. Langmuir and pseudo-second-order models indicated the monolayer adsorption and high capacity on the SF@Cu-NFs. Meanwhile, the adsorption thermodynamics showed that the spontaneous and exothermic process. As compared, SF@Cu-NFs indicated as an excellent adsorbent for Pb(II) treatment with the Q_max as high as 1908 mg g-1, which was about 3-20 folds than that of the other adsorbents.

By ascribing to its individual organic and inorganic component, the adsorption mechanism of SF@Cu-NFs for Pb(II) removal was discussed and revealed with two stages of fast adsorption and slow adsorption. On one hand, the flower ‘stamen’ of organic SF protein was designated as responsible adsorption site for fast adsorption of Pb(II). On the other hand, the flower ‘petal’ of inorganic Cu3(PO4)2 crystal was designated as responsible adsorption site for slow adsorption of Pb(II). This result clearly indicated that the silk fibroin protein-derived hybrid nanoflower could adsorb HMI Pb(II) well because of the adsorption site on the adsorbent surface.

In this work, we further understand the adsorption behavior and interaction process of HMI Pb(II) on the surface of silk fibroin derived hybrid nanoflowers. Through the present study, it has been successful to reveal the microscopic interaction process of Pb (II) adsorption that provides a new insight on understanding the adsorption mechanism. Also, based on interfacial adsorption, it is of great significance to comprehend the development of heavy metal ion removal applications. By fabricating SF@Cu-HNFs hybrid nanoflowers derived from SF protein, this work not only successfully provides insights on its adsorption performance and interaction mechanism for Pb(II) removal, but also significantly indicates its potential applications in contamination adsorption for environmental treatment.”.

The corresponding contents were added in the revised manuscript (Line942-966 in page 19).

References

  1. Chen, F., et al., Enhanced adsorption and photocatalytic degradation of high-concentration methylene blue on Ag2O-modified TiO2-based nanosheet. Chemical Engineering Journal, 2013, 221, 283-291.
  2. Saepurahman, M.A. Abdullah, and F.K. Chong, Dual-effects of adsorption and photodegradation of methylene blue by tungsten-loaded titanium dioxide. Chemical Engineering Journal, 2010,158(3),418-425.

Reviewer 2 Report

I do believe  that  the paper needs to be reorganized in a better format 

Firstly the introduction  despite the fact that is large  and  has 55 references needs to explain more clearly what is the novel character.  what is really new in the elaboration and characterization of SF@Cu-HNFs  ?
  The Pb removal was evaluated through  thermodynamic and dynamic investigation using Langmuir,  Freundlich  and  Temkin model establishing the well  fittings of Langmuir and pseudo-second-order models ( the monolayer adsorption ). The manuscript describes with details  the adsorption mechanism of SF@Cu-HNFs for Pb(II) removal  with  respect to its individual organic and inorganic component. It is interesting , but  In my opinion  the paper is not well organized having 11 figures and 1 table; may be a part of data from fig 7 could be introduce in a more complete table 1.

 May be it will be in the benefit of the manuscript as well to move a part of chapter conclusion in the section with discussion  

 In order to be published my recommendation is a moderate revision taking into account the above comments

Author Response

Response to Reviewer 2 Comments

Thank you for your letter and for the reviewers’ comments concerning our manuscript. Those comments are all valuable and very helpful for revising and improving our paper, as well as the important guiding significance to our researches. We have studied comments carefully and have made correction which we hope meet with approval. Revised portion are marked in red in the paper. The main corrections in the paper and the responds to the reviewer’s comments are as flowing:

Point 1: I do believe that the paper needs to be reorganized in a better format

Response 1:

     We have made correction according to the Reviewer’s comments.

     In fact, we first performed adsorption kinetics experiments to determine the adsorption equilibrium time, and then further conducted adsorption isotherm experiments and adsorption thermodynamic experiments. First, we reversed the order when we wrote the article. We have made correction according to the Reviewer’s comments. The corresponding contents were added in the revised manuscript (Line 184-213 in page 4).

     Second, we have carried out adsorption kinetics experiments. Secondly, the adsorption isotherms and thermodynamics experiments were carried out. At last, we did the selective adsorption experiment. The corresponding contents were restated in the revised manuscript (Line 484-761 in page 10-18).

    Thirdly, we have made correction according to the Reviewer’s comments and rewritten the conclusion.

Point 2: Firstly the introduction  despite the fact that is large  and  has 55 references needs to explain more clearly what is the novel character.  what is really new in the elaboration and characterization of SF@Cu-HNFs?

Response 2:

    First, heavy metal ions (HMIs) contamination has attracted widespread attention due to their highly toxicity and carcinogenicity. Pb(II) is one of the world’s leading environmental pollutants and has caused severe public health consequences. The adsorption technology is regarded as one of the most effective and competitive methods for Pb(II) treatment. As a result, the development of newly adsorbent material and the application to efficient Pb(II) removal are highly desirable for environmental protection. In this work, natural material of SF protein was used for the fabrication of protein-inorganic hybrid nanoflowers through self-assembly. SF protein is extracted from cocoon, which is cheap, easy to get and has a wide range of sources

    Second, the prepared Chinese peony flower-like SF@Cu-NFs products were applied to efficient adsorption of Pb(II). On one hand, the adsorption performance of SF@Cu-HNFs for Pb(II) removal was investigated and evaluated. Through thermodynamic investigation by Langmuir model and dynamic investigation by pseudo-second-order model, the adsorption of Pb(II) was indicated to be monolayer with high capacity on the SF@Cu-NFs surface. Meanwhile, the adsorption process was shown to be spontaneous and exothermic.

    Thirdly, the interaction mechanism of SF@Cu-HNFs for Pb(II) removal was discussed and revealed. The flower ‘stamen’ of organic SF protein was designated as responsible adsorption site for fast adsorption of Pb(II), which was originated from multiple and strong coordinative interaction produced between Pb(II) and abundant N, O elements on numerous amide groups. The flower ‘petal’ of inorganic Cu3(PO4)2 crystal was designated as responsible adsorption site for slow adsorption of Pb(II), which was originated from unique and weak coordinative interaction produced between Pb(II) and O element restricted from the strong ion bond by Cu(II) element. The different adsorption processes have been verified by intraparticle kinetic investigation and zeta potential measurement. By fabricating SF@Cu-HNFs hybrid nanoflowers derived from SF protein, this work not only successfully provides insights on its adsorption performance and interaction mechanism for Pb(II) removal, but also significantly indicates its potential applications in environment contamination treatment.

Point 3: The Pb removal was evaluated through thermodynamic and dynamic investigation using Langmuir, Freundlich and Temkin model establishing the well fittings of Langmuir and pseudo-second-order models (the monolayer adsorption). The manuscript describes with details the adsorption mechanism of SF@Cu-HNFs for Pb(II) removal with respect to its individual organic and inorganic component. It is interesting, but  In my opinion  the paper is not well organized having 11 figures and 1 table; may be a part of data from fig 7 could be introduce in a more complete table 1.

Response 3: We reversed the order and have made correction according to the Reviewer’ s comments. The corresponding contents were added in the revised manuscript (Line 184-213 in page 4). In order to shorten the expression of this part, we put the adsorption isotherms into the supplementary file (in page 15). The corresponding contents (Figure 7(b-c)) were removed in the revised manuscript.

Point 4: May be it will be in the benefit of the manuscript as well to move a part of chapter conclusion in the section with discussion 

Response 4: We have made correction according to the Reviewer’s comments and rewritten this part. Line 567-603 the statements were deleted and rewritten as following:

    In this work, natural material of SF protein was used for the fabrication of protein-inorganic hybrid nanoflowers through self-assembly and the three-dimensional structure was applied to efficient adsorption of HMI Pb(II).

    Through adsorption isotherms and kinetics, the adsorption performance of SF@Cu-HNFs for Pb(II) removal was systematically evaluated in detail. Langmuir and pseudo-second-order models indicated the monolayer adsorption and high capacity on the SF@Cu-NFs. Meanwhile, the adsorption thermodynamics showed that the spontaneous and exothermic process. As compared, SF@Cu-NFs indicated as an excellent adsorbent for Pb(II) treatment with the Q_max as high as 1908 mg g-1, which was about 3-20 folds than that of the other adsorbents.

    By ascribing to its individual organic and inorganic component, the adsorption mechanism of SF@Cu-NFs for Pb(II) removal was discussed and revealed with two stages of fast adsorption and slow adsorption. On one hand, the flower ‘stamen’ of organic SF protein was designated as responsible adsorption site for fast adsorption of Pb(II). On the other hand, the flower ‘petal’ of inorganic Cu3(PO4)2 crystal was designated as responsible adsorption site for slow adsorption of Pb(II). This result clearly indicated that the silk fibroin protein-derived hybrid nanoflower could adsorb HMI Pb(II) well because of the adsorption site on the adsorbent surface.

    In this work, we further understand the adsorption behavior and interaction process of HMI Pb(II) on the surface of silk fibroin derived hybrid nanoflowers. Through the present study, it has been successful to reveal the microscopic interaction process of Pb (II) adsorption that provides a new insight on understanding the adsorption mechanism. Also, based on interfacial adsorption, it is of great significance to comprehend the development of heavy metal ion removal applications. By fabricating SF@Cu-HNFs hybrid nanoflowers derived from SF protein, this work not only successfully provides insights on its adsorption performance and interaction mechanism for Pb(II) removal, but also significantly indicates its potential applications in contamination adsorption for environmental treatment.”.

The corresponding contents were added in the revised manuscript (Line942-966 in page 19).

Round 2

Reviewer 1 Report

The authors have performed the requested modifications on their manuscript, which now deserves for publication. I therefore recommend the publication of the manuscript. A last modification should be done (at the proofs stage for example) related to the point 4. The adsorption capacity is 2000 mg.g-1 and not 2000 g.mg-1 !